# Advances in Understanding Intestinal Homeostasis: Lessons from Inflammatory Bowel Disease and Monogenic Intestinal Disorder Pathogenesis

**DOI:** 10.3390/ijms26136133

**Published:** 2025-06-26

**Authors:** Céline Petit, Aurore Rozières, Gilles Boschetti, Christophe Viret, Mathias Faure, Stéphane Nancey, Rémi Duclaux-Loras

**Affiliations:** 1Centre International de Recherche en Infectiologie (CIRI), Université de Lyon, Université Lyon 1, Inserm U1111, CNRS UMR5308, ENS de Lyon, 69007 Lyon, France; 2Service d’Hépato-Gastroentérologie et Nutrition Intensive, Hôpital Lyon-Sud, Hospices Civils de Lyon, 69002 Lyon, France; 3Service de Gastroentérologie, Hépatologie et Nutrition Pédiatriques, Centre Constitutif des Maladies Rare Digestives, Hospices Civils de Lyon, Femme Mère-Enfant-Hospital, Boulevard Pinel, 69500 Bron, France

**Keywords:** inflammatory bowel disease, monogenic intestinal disease, very early-onset inflammatory bowel disease, inflammation, intestinal barrier, immune regulation

## Abstract

Inflammatory bowel diseases (IBDs) are chronic inflammatory conditions of the gastrointestinal tract that are multifactorial in nature. The pathophysiology involves interactions between the host immune system and environmental factors, including the gut microbiota, in genetically predisposed individuals. Advances in understanding these interactions have led to the development of novel therapeutic targets, ranging from anti-TNFα to more recent anti-interleukin 23 treatments. However, some patients still experience resistance to these therapies. Monogenic intestinal diseases (MIDs), which present with more severe symptoms than IBD and typically begin early in life, result from significant disruptions of intestinal homeostasis. MIDs are driven by mutations in a single gene, offering a unique opportunity to explore the mechanisms underlying intestinal homeostasis in health. In this review, we provide a comprehensive overview of the mechanisms of intestinal homeostasis by examining the cellular and molecular features of IBD and MID pathophysiologies.

## 1. Introduction

Inflammatory bowel disease (IBD) is a complex, multifactorial condition characterized by chronic inflammation of the gastrointestinal tract. It results from an excessive and inappropriate immune response to environmental factors—such as gut-microbiota-derived pathogens—in genetically predisposed individuals, ultimately leading to a disruption of intestinal homeostasis.

Advances in technologies such as single-cell sequencing, pangenomic studies, and microbiota analysis have significantly enhanced our understanding of both intestinal homeostasis and the pathophysiology of this multifactorial disease. Nevertheless, many aspects remain poorly understood.

In young children, the disruption of intestinal homeostasis can lead to severe, early-onset diseases such as congenital chronic diarrhea (CCD). Unlike IBD, CCD is often caused by mutations in a single gene, classifying it as a monogenic disease. The unique nature of these disorders provides a valuable model for studying the precise role of individual genes in maintaining intestinal homeostasis.

Over the past two decades, advances in next-generation sequencing have greatly facilitated the identification of genes associated with CCD [1,2,3]. To date, more than 100 genes have been implicated, shedding light on diverse molecular pathways essential for preserving gut integrity [3,4].

In this review, we will examine the common pathways shared between IBD and monogenic intestinal diseases (MIDs) (Table 1). Furthermore, we will explore how studying MID can not only enhance our understanding of IBD but also, in some cases, contribute to the development of targeted therapeutic strategies.

## 2. Altered Gastrointestinal Barrier in IBD and MID

The gastrointestinal (GI) mucosa, composed of a single layer of intestinal epithelial cells (IECs), is a complex multicellular structure with a dual role. It is crucial for both nutrient absorption and immune sensing, while also protecting the host from environmental antigens and microbial factors (Figure 1).

Intestinal homeostasis represents a delicate balance between nutrient absorption and protection against environmental factors. Multiple interconnected pathways involving various cell types within the intestine are crucial to maintaining this equilibrium and preventing nutritional deficiencies or disruptions of intestinal homeostasis. More than 70 monogenic diseases have been identified, each contributing to a breakdown in intestinal homeostasis by affecting essential mechanisms required to preserve the integrity of the mucosal barrier. Here, we compile the mutated genes known to play a vital role in intestinal homeostasis. These monogenic diseases can disrupt diverse pathways, including enterocyte function, mucus production, and both innate and adaptive intestinal immunity.

The intestinal barrier includes several distinct layers. The outer layer is represented by the mucus and the commensal gut microbiota. The mucus rests on a monolayer of specialized cells above the *lamina propria* containing immune cells. The monolayer comprises seven differentiated cell types: IECs, which absorb nutrients; goblet cells, which produce mucus; Paneth cells in the crypt, which specialize in the production of antimicrobial peptides (AMPs); enteroendocrine cells, which produce gut hormones; M cells, which manage the transcytosis of antigens from the luminal surface to the sub-epithelium; tuft cells, which monitor intestinal content; and stem cells, located at the base of the crypts, which drive epithelial renewal (Figure 1). The permeability of the monolayer is ensured by the junctional complex, which mainly consists of tight junctions (TJs), adherens junctions (AJs), and desmosomes. The disruption of the intestinal barrier in IBD and MID is associated with anomalies affecting the functionality of one or several of these cell types, leading to chronic intestinal inflammation.

### 2.1. Mucus and Goblet Cells

Mucus represents the first physical line of protection against the environment. Its multi-peptidoglycan composition allows for the retention of water and the formation of a dense glycocalyx layer. The structure of mucus varies from the small to the large intestine. The mucus layer in the small intestine is thinner, allowing for nutrient absorption, while in the colon, it is thicker and forms a double layer, providing stronger protection against pathogens. Additionally, goblet cells not only renew the mucus layer but also play an immune role by interacting with dendritic cells and secreting cytokines. Moreover, mucus production influences the composition of the microbiome. Twenty-two mucin proteins have been reported so far, among them a 5000 amino acid monomer rich in proline, serine, and threonine called MUC2, which is secreted by goblet cells [29]. In IBD patients, several defects in mucus function have been reported. Ulcerative colitis (UC) patients have a thinner mucus than Crohn’s disease (CD) patients with the excretion of less MUC2 during flare-ups [30,31,32]. Moreover, IBD patients show anomalies in the viscoelastic function of mucus due to higher levels of sulfide, a product of sulfate-reducing bacteria, which reduces the disulfide bonds within the mucus and breaks down the mucus network [12,33].

The post-translational processing of mucus is a mandatory step in acquiring its chemical properties. For example, sialylation, performed by the ST6GALNAC1 (ST6) enzyme, confers upon mucus resistance to bacterial degradation. Patients with biallelic deleterious variants in *ST6* develop VEOIBD, which is linked to a defect in ST6 activity (Figure 1) [12]. Moreover, the glycosylation pattern has been evaluated on sigmoid biopsies from UC patients and showed decreased O-glycosylation of MUC2 in patients with active colitis [33,34]. Mice lacking O-glycan synthesis develop spontaneous colitis, with decreased MUC2 expression and increased intestinal permeability [34].

Another monogenic disease associated with a defect in mucus production was recently reported in two consanguineous siblings presenting with VEOIBD without immunodeficiency. Whole exome sequencing (WES) pinpointed homozygous mutations in the anterior gradient 2 (AGR2) gene. AGR2 is an endoplasmic reticulum protein disulfide isomerase family member involved in the folding of proteins, including MUC2 [13]. Therefore, patients with AR *AGR2* mutations showed a depletion of MUC2. The patient’s histologic analysis showed extensive intestinal metaplasia with a loss of goblet cells on gastric and rectal biopsies, respectively. Intestinal inflammation was related to an increase in BiP expression, a marker of endoplasmic reticulum stress [13].

### 2.2. Enterocytes

The impaired permeability of the GI barrier is a fundamental aspect of IBD pathogenesis [35,36,37]. Gut biopsies from IBD patients have shown a downregulation of TJ complexes such as E-cadherin and β-catenin [38]. Moreover, in vivo studies using endomicroscopy have reported that the loss of IECs’ integrity preceded clinical symptoms in IBD patients [39].

Junction Defect

The epithelial cell adhesion molecule (EpCAM) is a protein responsible for maintaining IEC integrity by recruiting proteins of the claudin family, such as E-cadherin [40]. Patients with autosomal recessive (AR) *EpCAM* mutations exhibit severe intractable chronic diarrhea, necessitating long-term parenteral nutrition, a condition known as congenital tufting enteropathy (CTE) [41] (Figure 1). Indeed, intestinal biopsies from these patients reveal villous atrophy, crypt abnormalities, and intestinal epithelial cell tufts [42]. Interestingly, some of these patients also suffer from IBD-like intestinal inflammation [5].

The impact of TNF-α on TJs has been extensively studied. In vitro studies have shown a decrease in the number of tight junctions between epithelial cells [43]. Additionally, TNF-α has been implicated in the inhibition of occludin promoter activity, leading to the redistribution of occludin, ZO-1, and claudin-1. In patients with ulcerative colitis (UC), several studies have demonstrated a predominance of claudin-2 expression in colonic biopsies, accompanied by a concomitant decrease in other TJ proteins, including reduced staining of claudins 3, 4, and 7 [44]. Similarly, in Crohn’s disease (CD) patients, claudin-2 appears to be increased in colonic inflammatory tissue. However, some studies have reported the opposite findings, showing decreased levels of claudins 2 and 12 in sigmoid tissues [45].

Transport Defect

Congenital chloride diarrhea, first described in 1945, is related to a loss-of-function (LOF) mutation in the solute carrier family 26 member 3 (SLC26A3). Patients present with severe neonatal diarrhea associated with a high chloride concentration in their stools due to a defect in the chloride [Cl^−^]/bicarbonate [HCO_3_^−^] exchanger SLC26A3, located at the brush border of IECs. Norsa et al. reported IBD-like intestinal inflammation in 17% of a cohort of 72 patients [6]. Interestingly, a genome-wide association study identified a polymorphism on SLC26A3 in a UC Japanese cohort [46]. The link between anionic transport defect and the disruption of the gastrointestinal barrier may be due to the change in salt concentration, which could impact the microbiota composition by promoting colitogenic bacteria and increase the patient’s susceptibility to developing intestinal inflammation [47,48]. Similarly, congenital sodium diarrhea is linked to a defect in SLC9A3, which encodes the Na^+^/H^+^ antiporter 3 (NHE3). A cohort of nine patients reported in 2015 included two individuals who developed inflammatory bowel disease (IBD) at 4 and 16 years of age [49]. Moreover, genome-wide association studies (GWASs) in IBD patients have shown a strong association between ulcerative colitis and the SLC9A3 locus. NHE3 activity is also reduced in both ulcerative colitis and Crohn’s disease, particularly in areas of active colitis and inflammation [50].

Enterocyte Architecture

The intestinal epithelial monolayer is mainly formed by IECs. IECs are polarized cells with an apical pole in contact with the intestinal lumen and a basal pole in contact with the lamina propria. To increase the absorption surface, IECs possess 3D microvillous actin protrusions located on their apical pole. This structure, composed of actin filaments anchored to the terminal web (a complex protein platform in the cytoplasm), plays a crucial role in nutrient absorption [42]. Actin structures are attached to the plasma membrane through transmembrane proteins, such as the ezrin/radixin/moesin (ERM) complex. Intracytoplasmic vesicles are necessary for the transport of ezrin proteins to the apical pole of the enterocytes. Inborn errors of myosin 5b (MYO5B), a gene coding for an enterocyte vesicular motor protein that interacts with Rab proteins, lead to microvillus inclusion disease (MVID), characterized by the absence of microvillus formation, which results in severe neonatal secretory diarrhea (Figure 1) [10,51]. Moreover, MYO5B KO mice exhibit an aberrant localization of transporters such as sodium–hydrogen exchanger 3 (NHE3) [52]. Although intestinal inflammation is not the clinical hallmark of MYO5B deficiency, some studies report patients with colitis or upper GI inflammation [53]. It is plausible that the aberrant localization of key transporters on the brush border may modify the ionic environment in the GI lumen, which could be responsible for the enrichment of proinflammatory microbiota. However, few functional validation experiments are available in the literature. Recent in-depth intracellular functional evaluations of different MYO5B patient mutations have demonstrated a correlation between genotype and phenotype [52]. Further evidence proving the causal relationship between specific intracellular dysfunctions and distinct mutations could improve our understanding of patient phenotypes and the presence or absence of inflammation. Several mutations of other genes have been described as altering enterocyte architecture, such as STX3, coding for a docking vesicle protein [54], or UNC45A, a gene coding for a myosin co-chaperone protein [55]. Yet, inflammation has not been associated with these diseases. A recent study examining single-nucleotide variants (SNVs) associated with very early-onset IBD (VEOIBD) susceptibility found a significant enrichment of MYO5B variants in this population [56].

Several other myosins are involved in IEC functions. Polymorphisms in MYO9B, involved in cytoskeleton remodeling, have also been reported to be linked to IBD susceptibility [57]. Myosin light-chain kinase (MLCK) is a key protein for enterocyte detachment [58]. Moreover, MLCK has been reported to be overactivated in IBD patients’ gut biopsies, which may result in altered epithelium integrity [59]. Interestingly, the phosphorylation of myosin 2, through MLCK, is regulated by the RhoA/ROCK pathway, a protein complex involved in the rearrangement of the cytoskeleton and cell dynamics. Rocío López-Posadas et al. showed that Rho-A in IECs accumulated in the cytosol of biopsies from IBD patients [60]. Moreover, the in vitro activation of Rho-A was shown to promote an increased production of nuclear factor kappa B (NFkB) [11]. So far, no monogenic human diseases related to RhoA/ROCK have been reported. However, it has been demonstrated that organoids derived from the gut biopsies of tetratricopeptide repeat domain 7A (TTC7A) loss-of-function (LOF) patients presented a hyperphosphorylation of myosin 2 related to the overstimulation of the RhoA/ROCK pathway. Patients present multiple neonatal intestinal atresia with intestinal inflammation associated with immunodeficiency, such as hypogammaglobulinemia. TTC7A is a scaffolding protein that recruits phosphatidylinositol 4-kinase III alpha, which is important for synthesizing PI4-phosphate (PI4P) [61]. AR mutations in TTC7A influence the formation of PI4P, which subsequently regulates the RhoA/ROCK pathway. In vitro, organoids derived from patient biopsies showed an inverted apicobasal expression of actin [62]. Of note, treatment of TTC7A-deficient organoids with the ROCK inhibitor Y-27632 restored the organoids’ polarity (Figure 2).

Overall, the literature highlights several defects in the intestinal barrier in IBD pathophysiology, with particular insights provided by MID.

The phosphoinositide PI4P serves as a precursor for multiple signaling lipids involved in epithelial polarity. PI4P can be phosphorylated by PIK3C2A to generate PI(3,4)P_2_, a key regulator of intestinal epithelial cell polarity. In parallel, PI4P can be converted into PI(4,5)P_2_ by PI5K, and subsequently into PI(3,4,5)P_3_ through the action of PI3K, highlighting the central role of PI4P-derived lipids in membrane identity and cellular signaling.

PI(3,4,5)P_3_ acts as a membrane docking site for the serine/threonine kinase AKT, promoting its recruitment and activation. Once activated, AKT supports intestinal epithelial cell survival through downstream signaling involving mTOR and the anti-apoptotic protein BCL2. Additionally, AKT inhibits pro-apoptotic factors such as BAD, FOXO transcription factors, and p53, thereby limiting apoptosis.

Upon activation, AKT inhibits the activation of RhoA by preventing the exchange of GDP for GTP, thereby impairing the activation of the Rho-associated kinase (ROCK) pathway. ROCK, in turn, plays a crucial role in regulating actomyosin contractility by inhibiting the dephosphorylation of myosin light chain 2 (MLC-2). By inhibiting ROCK activity, AKT effectively downregulates actomyosin contraction, demonstrating a negative regulatory effect on cellular contractility.

TTC7A deficiency disrupts these signaling pathways at multiple levels. Impaired PI4P metabolism leads to a failure in the conversion to PI(3,4)P_2_ and PI(3,4,5)P_3_, reducing AKT activation and compromising epithelial cell survival. As a result, there is an increase in apoptosis and a decrease in cell survival. Additionally, TTC7A deficiency causes the deregulation of the ROCK pathway due to impaired AKT-mediated inhibition of RhoA, resulting in aberrant actomyosin contraction. In addition to the disruption of epithelial cell signaling, TTC7A deficiency is associated with intestinal dysbiosis and an increased abundance of enterobacteria. Due to the loss of epithelial cell polarity, these bacteria can more easily invade the lamina propria, where they are recognized by dendritic cells. The dendritic cells then present the bacteria to naïve T lymphocytes, which differentiate into effector T cells. These effector T cells produce pro-inflammatory cytokines, contributing to the inflammatory response observed in these patients. Furthermore, naïve T cells can differentiate into follicular helper T cells, which support the differentiation of B cells into plasma cells that produce antibodies. However, in TTC7A-deficient patients, this immune response is impaired, leading to a lack of antibody production and resulting in hypogammaglobulinemia. Additionally, studies have shown that lymphocyte migration is altered in these patients, further complicating the immune response.

Finally, TTC7A deficiency also results in a pseudostratified epithelium, a feature that may be attributed to the loss of polarity in the enterocytes, further contributing to epithelial dysfunction.

## 3. Altered GI Microbiota–Host Mucosal Interaction in IBD and MID

Beyond the physical separation of the host from the intestinal microbiota, intestinal epithelial cells (IECs) and immune cells employ multiple defense mechanisms to regulate microbial composition and preserve mucosal homeostasis.

### 3.1. Innate Immunity in Epithelial Cells

GUCY2C

Guanylate cyclase C (GC-C) is a transmembrane receptor predominantly expressed in intestinal epithelial cells (IECs). Upon activation by heat-stable enterotoxins derived from *Escherichia coli*, GC-C catalyzes the formation of cyclic monophosphate (cGMP). This increase in intracellular cGMP subsequently stimulates the cystic fibrosis transmembrane conductance regulator (CFTR) and inhibits sodium-hydrogen exchanger 3 (NHE3), leading to water secretion into the intestinal lumen [7]. Additionally, cGMP signaling is linked to the regulation of inflammation and cell proliferation [8].

In pediatric populations, GUCY2C dysfunction manifests in two primary phenotypes. Homozygous or compound heterozygous loss-of-function (LOF) mutations in GUCY2C result in meconium ileus, while gain-of-function (GOF) mutations—characterized by excessive cGMP production—lead to severe neonatal diarrhea, dependence on total parenteral nutrition, and an inflammatory bowel disease (IBD)-like phenotype [8,9,63].

The contribution of impaired sodium transport via GC-C to IBD pathogenesis remains contentious. Some studies have observed elevated rectal mucosal cGMP levels in ulcerative colitis (UC) patients [64], whereas others have reported GC-C downregulation in both UC and CD [65]. GC-C is also implicated in maintain intestinal barrier integrity through tight junctions [66].

ALPI

Mucosal immune regulation is partly maintained by the IEC-mediated detection of microbial components. Toll-like receptors (TLRs), a class of pattern recognition receptors, detect microbial ligands such as lipopolysaccharide (LPS)—a key component of Gram-negative bacterial membranes. The activation of TLR4 by LPS induces the nuclear factor kappa B (NF-κB) signaling cascade and subsequent pro-inflammatory responses [67,68].

Intestinal alkaline phosphatase (ALPI), a brush border enzyme, dephosphorylates LPS, thereby attenuating TLR4 activation. AR mutations in ALPI have been reported in VEOIBD [14]. In vitro studies using HEK cells expressing mutated ALP1 from affected individuals demonstrated reduced LPS dephosphorylation activity. Clinically, these patients often exhibit autoimmune features such as autoantibody production and concurrent celiac disease. In ALPI expression is significantly reduced in the terminal ileum and colonic biopsies from patients with active CD and UC, respectively [69]. Moreover, in pediatric patients with active IBD, ALPI expression was markedly lower in inflamed tissue compared to controls [70].

DUOX2/NOX1

The regulation of the gut microbiota is also mediated by the production of reactive oxygen species (ROS) via epithelial NADPH oxidases—dual oxidase 2 (DUOX2) and NADPH oxidase 1 (NOX1). These enzymes generate hydrogen peroxide (H_2_O_2_) within the intestinal lumen, contributing to antimicrobial defense [71,72]. AR LOF mutations in DUOX2 were initially identified in children with congenital hypothyroidism [73]. Subsequent WGS revealed missense mutations in DUOX2 in individuals with VEOIBD. A functional assay demonstrated reduced ROS production compared with controls [15]. A comprehensive multiomic study further associated rare LOF DUOX2 variants with elevated plasma levels of proinflammatory IL-17C [71]. NOX1 missense mutations have also been linked to VEOIBD [15].

In IBD patients, DUOX2 expression is increased in the colon. Moreover, CD patients show elevated DUOX2 mRNA expression in non-inflamed biopsies [74]. However, excessive DUOX2 expression in the crypt epithelial cells of IBD patients has been reported, which may be detrimental to intestinal stem cells [75]. The precise roles of DUOX2 and NOX1 in the pathophysiology of IBD remain under investigation.

### 3.2. Innate Immunity in Immune Cells

Neutrophils’ and Macrophages’ Phagocytosis: NADPH Oxidase

The hallmark of active IBD is neutrophilic infiltration within the intestinal mucosa. Neutrophils, as first responders to inflammation, combat microbial invasion through ROS production. Their role in IBD, however, is complex and context-dependent, with subsets exhibiting both pro-inflammatory and protective effects [76]. Rodent models have yielded contradictory results regarding anti-neutrophil therapies, with outcomes ranging from the amelioration to the exacerbation of colitis [77,78,79]. In UC patients, neutrophils exhibit elevated markers for neutrophil extracellular traps (NETs) such as peptidylarginine deiminase 4 (PAD4), myeloperoxidase (MPO), and citrullinated histone H3 (citH3) [80]. Moreover, neutrophils from IBD patients release more NETs that localize in the colon, and NET degradation is impaired compared to neutrophils from healthy subjects [80]. In contrast, higher receptors of neutrophil endothelial transmigration such as CD177 are expressed in IBD patients. CD177-positive neutrophils, which are enriched in IBD, produce elevated ROS, antimicrobial peptides (AMPs), and NETs, while displaying a diminished secretion of pro-inflammatory cytokines (IL-6, IL-17A, and IFN-γ) and increased levels of regulatory cytokines (TGF-β and IL-22), facilitating mucosal repair [81,82]. Recently, antibodies against granulocyte-macrophage colony-stimulating factor (GM-CSF) were reported to predict severe ileal CD before diagnosis [83]. These antibodies decreased the abundance of myeloid cells, notably neutrophils, and increased group 1 innate lymphoid cells, leading to disrupted intestinal homeostasis.

AR mutations in NADPH oxidase mutations result in chronic granulomatous disease (CGD), characterized by severe bacterial and fungal infections leading to repetitive abscesses associated with hyperinflammation, such as colitis with perianal lesions [16]. Hematopoietic stem cell transplantation (HSCT) remains the only curative therapy. NADPH oxidase is a multi-structure protein, which plays several roles in neutrophil defense against pathogens. It generates superoxide anions leading to antimicrobial oxidants such as hydrogen peroxide and hypohalous acid. NADPH oxidase activation leads to the activation of antimicrobial proteases stored in the granules of neutrophils [84]. The NADPH oxidase complex also contributes to NET generation, which requires further exploration to better understand its role. The NADPH oxidase protein complex is composed of transmembrane proteins, gp91phox (phagocyte oxidase) and p22phox, and cytoplasmic proteins, p47phox, p67phox, and p40phox. One-third of CGD cases are X-linked due to CYBB gene mutations coding for gp91phox, and autosomal recessive cases involve CYBA, NCF1, NCF2, or NCF4, coding, respectively, for p22phox, p47phox, p67phox, and p40phox (Figure 2). Defective NADPH oxidase impairs neutrophil bactericidal activity, resulting in immune deficiency and chronic inflammation. In CGD-associated IBD, autoinflammatory mechanisms predominate, driven by impaired ROS production, defective autophagy, and elevated IL-1 secretion [85]. Notably, the impaired ability of neutrophils to clear commensal microbes does not necessarily result in overt infection but perpetuates mucosal immune dysregulation. Assessment of oxidative burst from neutrophils in suspected CGD patients remains the gold standard for CGD diagnosis [86].

### 3.3. Innate Immunity in Epithelial Cells and Immune Cells

NOD2/CARD15

Mutations in nucleotide-binding oligomerization domain 2 (NOD2), also known as caspase recruitment domain-containing protein 15 (CARD15), were the first identified gene alterations conferring susceptibility to Crohn’s disease [87]. To date, over 60 polymorphisms in this gene have been identified [88]. NOD2 is a key component of the innate immune system, serving as an intracellular pattern recognition receptor (PRR) that detects muramyl dipeptide (MDP), a conserved motif within bacterial peptidoglycan (Figure 3). NOD2 activates pro-inflammatory pathways and other innate immune pathways, including autophagy and endoplasmic reticulum stress. The interaction between NOD2 and MDP triggers the NF-kB signaling cascade, a pro-inflammatory transcription factor. The NOD2 pathway is balanced between activator and inhibitor proteins, resulting in the regulation of NF-kB [89]. In Crohn’s disease, NOD2 mutations disrupt the negative regulation of the NOD2–NF-κB signaling axis, leading to excessive immune activation. These mutations are also associated with decreased interleukin-10 (IL-10) production, potentially due to the impaired activity of heterogeneous nuclear ribonucleoprotein A1 (hnRNP A1). Although no monogenic intestinal diseases directly caused by NOD2 mutations have been identified, both AR and AD mutations in downstream regulators of the NOD2 cascade have been described. NOD2-mediated NF-κB activation requires interaction with receptor-interacting protein kinase 2 (RIPK2), which must be ubiquitinated by the X-linked inhibitor of apoptosis protein (XIAP). XIAP is a multidomain E3 ubiquitin ligase whose RING domain facilitates ubiquination, enabling NF-κB signaling. A deficiency in XIAP, which is linked to X-linked conditions, can lead to hemophagocytic lymphohistiocytosis (HLH), often triggered by Epstein–Barr virus (EBV) infections [90]. Notably, XIAP-deficient patients develop severe colitis in 25% of cases, likely due to impaired microbial clearance by intestinal and immune cells, leading subsequently to uncontrolled inflammation [91].

NADPH oxidase is a multi-structure protein, which plays several roles in neutrophil defense against pathogens. It generates superoxide anions, leading to antimicrobial oxidants such as hydrogen peroxide and hypohalous acid. It is composed of transmembrane proteins, gp91phox (phagocyte oxidase) and p22phox, and cytoplasmic proteins, p47phox, p67phox, and p40phox. One-third of CGD cases are X-linked due to CYBB gene mutations coding for gp91phox, and autosomal recessive cases involve CYBA, NCF1, NCF2, or NCF4, coding, respectively, for p22phox, p47phox, p67phox, and p40phox.

NOD2 is a key component of the innate immune system, playing a crucial role in detecting and responding to pathogens. This intracellular receptor specifically recognizes muramyl dipeptide (MDP). The interaction between NOD2 and MDP triggers the NF-kB signaling cascade. The NOD2 pathway is balanced between activator and inhibitor proteins, resulting in the regulation of NF-kB. The activation of the NF-κB pathway is initiated when NOD2 interacts with receptor-interacting protein kinase 2 (RIPK2). For this activation, RIPK2 must be ubiquitinated by the X-linked inhibitor of apoptosis (XIAP) protein. XIAP, a multidomain protein with an RING domain that functions as an E3 ubiquitin ligase, targets RIPK2 to promote the NF-κB pathway. Mirroring XIAP ubiquitin function, ITCH (AIP4), an HECT domain containing E3 ubiquitin ligase, is important for RIPK2 ubiquitination. Once NOD2 interacts with RIPK2, the complex recruits transforming growth factor-β activated kinase 1 (TAK1), which facilitates the ubiquitination and degradation of inhibitor of kappaB kinase (IKKγ), also named NF-kB essential modulator (NEMO). This process leads to the phosphorylation of IκBα by the IKKα and IKKβ subunits, ultimately activating NFκB. TNFα-induced protein 3 (TNFAIP3), also known as A20, regulates NEMO activities.

In Paneth cells, NOD2 signaling normally promotes the secretion of chemokines such as IL-8 and MCP-1, reducing the chemoattraction of granulocytes important for microbial clearance. Similar to XIAP, the E3 ubiquitin ligase ITCH (AIP4), which contains an HECT domain, also mediates RIPK2 ubiquitination [92] (Figure 4). ITCH deficiency is associated with facial dysmorphia and auto-inflammatory IBD-like mucosal disease [17,93].

Following NOD2-RIPK2 interaction, the complex recruits transforming growth factor-β activated kinase 1 (TAK1), which drives the ubiquitination and degradation of inhibitor of kappaB kinase (IKKγ), also named NF-kB essential modulator (NEMO) (Figure 4). This leads to the phosphorylation of IκBα by the IKKα and IKKβ subunits, ultimately activating NFκB [94]. Mutations in IKBKG, the gene encoding NEMO, result in severe colitis, ectodermal dysplasia, and immunodeficiency, including hypogammaglobulinemia [18]. Additionally, dysfunctions in the binding of TNFα-induced protein 3 (TNFAIP3), also known as A20, a regulator of NEMO, can lead to bowel inflammation, arthritis, and oral and genital ulceration similar to Behçet disease, associated with immunodeficiency [19].

Inflammasomes

Inflammasomes are cytosolic multiprotein complexes that regulate the activation of proinflammatory caspases in response to microorganisms. They are triggered by pathogen-associated molecular patterns (PAMPs) or danger-associated molecular patterns (DAMPs). Canonical inflammasome activation involves sensors such as NLRP3 (NOD-like receptor family, pyrin domain containing 3) or AIM2 (absent in melanoma 2), which recruit apoptosis-associated speck-like protein containing a CARD (ASC) and pro-caspase-1. Once assembled, the complex activates caspase-1, which cleaves pro-IL-1β and pro-IL-18 into their mature forms, and processes gasdermin D, whose pore-forming activity facilitates cytokine release. Non-canonical inflammasomes, through caspase-4/5 activation, also cleave gasdermin D, contributing to pyroptosis and cytokine release. Inflammasomes such as NLRP1, NLRP3, NLRC4, AIM2, and Pyrin are pivotal in maintaining the intestinal immune equilibrium.

Despite elevated IL-1β levels observed in Crohn’s disease, the precise role of inflammasomes—particularly NLRP3—remains controversial. Confounding factors include genetic heterogeneity, microbial composition, and variability in experimental colitis models.

Monogenic inflammasomopathies offer valuable insights into intestinal immune regulation. For instance, Zhou et al. showed that gain-of-function (GOF) mutations in NLRP3 (e.g., R779C) that were identified in very early-onset IBD (VEOIBD) enhanced NLRP3 deubiquitination. Similarly, mutations in NLRC4 are associated with AD disorders featuring poor growth and gastrointestinal symptoms in infancy [95]. In a related manner, Pyrin, a cytosolic protein encoded by the MEFV gene, is crucial for inflammasome activation. Mutations in MEFV cause familial Mediterranean fever (FMF), a common hereditary autoinflammatory syndrome. Studies have shown a higher prevalence of intestinal inflammation among FMF patients compared to the general population.

Autophagy

Pangenomic studies in IBD patients have identified several polymorphisms, including variants in genes involved in autophagy, such as the ATG16L1 T300A mutation. Autophagy is a conserved, multi-step process essential for cellular homeostasis, encompassing the degradation of misfolded proteins, damaged organelles, and intracellular pathogens (xenophagy). Autophagic cargo is sequestered via membrane nucleation (initiation), expansion (elongation), and the formation of autophagosomes, which subsequently fuse with lysosomes to form autolysosomes for degradation (maturation). This entire sequence is termed autophagic flux.

The ATG16L1 T300A variant impairs the clearance of adherent-invasive Escherichia coli (AIEC)—a bacterium enriched in Crohn’s disease—highlighting a key role for autophagy in microbial defense [96]. Additionally, Cadwell et al. have demonstrated the importance of ATG16L1 in the formation of lysozyme granules within Paneth cells in intestinal crypts in both mice and humans, as well as the impact of the *ATG16L1T* 300A variant on lysozyme secretion and the promotion of norovirus-induced colitis in a murine model [97,98]. More broadly, autophagy regulates immunity by degrading inflammatory factors and shaping the repertoire of antigenic peptides presented by major histocompatibility complex (MHC) class II molecules to T cells. These functions are particularly critical to the role of dendritic cells (DCs), which are key players in intestinal homeostasis. Furthermore, dysfunctional DC activity has been associated with several inflammatory diseases, including Crohn’s disease. Recently, our team identified a major defect in the adaptation of the autophagic flux during the maturation of dendritic cells (DCs) from adult patients with Crohn’s disease homozygous for the *ATG16L1T* 300A variant [99].

Other autophagy genes associated with CD susceptibility include ULK1, ATG4C, ATG9A, and NDP52 [100,101,102]. Also associated with CD risk are genes that attenuate inflammation by promoting autophagy-dependent inflammasome regulation. This occurs downstream of bacterial entry in the case of NOD2 [103,104] or at the level of inflammasome assembly in the case of IRGM1 [105,106].

These findings also suggest that impaired autophagic activity in patients with Crohn’s disease could ultimately contribute to a disruption of intestinal homeostasis, promoting digestive inflammation.

Pediatric monogenic diseases involving the autophagic process have been primarily reported in neurological conditions, such as mutations in *WDR45*, which are responsible for neurodegeneration with brain iron accumulation. However, no monogenic intestinal diseases involving the autophagic process have been documented in the literature to date. Although no monogenic intestinal diseases have been directly linked to core autophagy genes, several monogenic disorders—including XIAP and NPC1 deficiencies—feature Crohn’s-like colitis and involve impaired NOD2-induced autophagy. NOD2, the strongest genetic risk factor for Crohn’s disease, interacts with ATG16L1 and is crucial for antimicrobial autophagy in phagocytes. Therapies targeting this pathway, such as the mTOR inhibitor rapamycin, have shown promise in restoring autophagy and bacterial clearance in NOD2-variant cells.

### 3.4. Enteroencrine and Neuroendocrine Cells

Enteroendocrine cells (EECs) are specialized epithelial cells scattered throughout the gastrointestinal tract that sense luminal nutrients and secrete hormones in response. These hormones, such as GLP-1, GIP, and CCK, regulate various aspects of digestion, glucose metabolism, and appetite. EECs function as key intermediaries between the gut lumen and systemic physiological responses, integrating signals to maintain gastrointestinal and metabolic homeostasis. Several MIDs have been linked to EEC dysfunction, involving genes such as NEUROG3, PERCC1, and RFX6. Mutations in these genes cause chronic malabsorptive diarrhea, often associated with endocrinologic abnormalities—for example, early-onset diabetes in NEUROG3 deficiency. However, to date, these patients have not been reported to develop intestinal inflammation. Luis F. Sifuentes-Dominguez et al. reported mutations in SCGN, a gene that encodes Secretagogin, a calcium-sensing protein exclusively expressed in neuroendocrine cells, including EECs [107]. In this study, three related patients presented with early and severe intestinal inflammation. Secretagogin interacts with the SNARE complex and regulates hormone secretion [108,109]. This example illustrates a potential link between the neuroendocrine system and intestinal immunity, highlighting the importance of neuro–gut interactions in maintaining intestinal homeostasis.

### 3.5. Microbiota in IBD and MID

The gut microbiota plays a crucial role in the pathogenesis of inflammatory bowel disease (IBD) by modulating immune responses and driving chronic inflammation. Dysbiosis in IBD is characterized by a decrease in beneficial bacteria such as *Faecalibacterium prausnitzii* and an increase in pro-inflammatory species like *Escherichia coli* and *Bacteroides fragilis*. This imbalance leads to the disruption of the intestinal barrier, allowing microbial-derived antigens, including lipopolysaccharides (LPS) and flagellin, to activate pattern recognition receptors (e.g., TLR4 and NOD2) on antigen-presenting cells. This activation triggers the excessive production of pro-inflammatory cytokines such as TNF-α, IL-6, and IL-23, promoting the differentiation of Th17 and Th1 cells, which further release IL-17 and IFN-γ, respectively, amplifying chronic intestinal inflammation. Moreover, a reduction in short-chain fatty acid (SCFA)-producing bacteria impairs regulatory T cell (Treg) function, reducing IL-10 production and exacerbating the inflammatory response [110]. This intricate interplay between microbial dysbiosis and the immune system is central to IBD pathophysiology.

The gut microbiota colonizes and develops primarily between birth and three years of age—a period that coincides with the typical age of onset of very early-onset inflammatory bowel disease (VEO-IBD). Sokol et al. compared the fecal microbiota of patients with VEO-IBD and those with monogenic immune disorders (MIDs) such as chronic granulomatous disease (CGD), XIAP deficiency, and TTC7A deficiency [111]. They highlighted distinct microbial signatures across these conditions, including a higher abundance of *Bacteroidetes* in CGD, an increase in *Proteobacteria* in TTC7A deficiency, and elevated levels of *Clostridiaceae* in XIAP deficiency.

Defects in immune regulation are also linked to dysbiosis. Foxp3 mutant scurfy (SF) mice, a model of human IPEX syndrome, exhibit an autoimmune phenotype even under germ-free conditions, due to the loss of regulatory T cell (Treg)-mediated control of inflammatory effector T cells. Dysbiosis has also been observed in FOXP3-DTR mice and in mice lacking B cells (*Ighm*−/−), T cells (*Cd3e*−/−), or both B and T cells (*Rag1*−/−) [112]. Additionally, children with severe atopic dermatitis have been found to harbor altered fungal skin microbiomes, with increased colonization by *Clostridium* species, *Serratia marcescens*, and opportunistic fungi such as *Candida* and *Aspergillus*. However, further functional validation studies are needed to elucidate the precise role of individual mutations in the development of dysbiosis.

## 4. Altered Adaptive Immunity in IBD and MID

The balance between the inflammatory response and non-inflammatory response is crucial for maintaining intestinal homeostasis. Beyond the important role of innate immune mechanisms in intestinal homeostasis maintenance, the adaptive immune system—comprising effector T cells and B cells and located in Peyer’s patches and the lamina propria of the intestinal mucosa—is also a cornerstone of host responses to environmental and gut microbiota stimuli.

### 4.1. T-Effectors: TH1/TH2/TH17

Naïve T cells differentiate into distinct effector subsets based on their microenvironment. Key cytokines, including IL-12, IL-4, IL-6, and TGF-β, influence this differentiation by activating specific transcription factors. For instance, IL-12 induces T-bet, promoting Th1 cell development and IFN-γ production; IL-4 activates GATA3, leading to Th2 cell differentiation and IL-4, IL-5, and IL-13 secretion; and IL-6 and TGF-β stimulate STAT3, driving Th17 cell differentiation and the production of IL-17 and IL-22 (Figure 1).

IBD is characterized by an increased presence of activated CD4+ T cells in the intestinal lamina propria [113,114]. While CD is predominantly associated with a Th1 and Th17 cell-mediated immune response [115], UC is linked to Th2-driven inflammation. Elevated levels of Th1-associated cytokines (IFN-γ, TNF-α, IL-12, and IL-23) have been observed in both the serum and mucosal tissues of IBD patients [116,117,118]. Conversely, Th2 cytokines (IL-13 and IL-5) are more prominent in UC.

Th17 cells play a complex role in IBD. Although they contribute to mucosal defense against pathogens, they can also drive inflammation. The differentiation into pro- or anti-inflammatory subsets depends on the cytokine milieu: IL-6, IL-23, and IL-1β promote a pathogenic Th17 phenotype, whereas TGF-β and IL-6 induce a regulatory phenotype. Th17 cells contribute to intestinal inflammation by producing pro-inflammatory cytokines, recruiting neutrophils, and modifying intestinal permeability [119]. Targeting the shared p40 subunit of IL-12 and IL-23 with biotherapies has been shown to decrease disease activity in CD and UC [120,121]. Moreover, recent advances have demonstrated that rizankizumab, which targets the p19 subunit of IL-23, is more effective than anti-p40 [122].

From the perspective of monogenic diseases, numerous innate immunodeficiency syndromes have been associated with intestinal inflammation. Some affect a broad range of components of the adaptive immune system, while others are more specific. Inducible T cell co-stimulator (ICOS) deficiency is a condition in which the ICOS receptor, which is expressed on the surface of lymphocytes, is impaired. This receptor mediates cell–cell interactions and allows for the transduction of secondary signals necessary for lymphocyte activation. ICOS supports the survival, proliferation, differentiation, and regulation of Th1, Th2, and Th17 cells and is essential for IL-10 production and immunoglobulin synthesis. Patients with ICOS deficiency develop common variable immunodeficiency (CVID) type 1, characterized by hypogammaglobulinemia, severe enteropathies, autoimmunity, lymphoproliferation, and malignancies. Studies have shown that lymphocytes from ICOS-deficient patients exhibit a higher Th1 response than controls following LPS/IFNγ stimulation. Ustekinumab, a biologic agent targeting the p40 subunit of IL-12/IL-23, has been identified as a potential treatment for ICOS deficiency-related IBD-like disease [20].

The dedicator of cytokinesis 8 (*DOCK8*) gene is crucial for immune responses influencing both actin cytoskeleton-dependent and -independent immune responses. It facilitates the accumulation of adhesion molecules and cytotoxic granules at immunologic synapses, essential for B, T, and NKT cell survival. DOCK8 deficiency leads to cytothripsis, decreased immunoglobulin levels (IgG, IgA, and IgM), elevated IgE (associated with allergic reactions), reduced peripheral naïve T cells, and an increased proportion of activated T cells producing TH2 cytokines [21]. Additionally, DOCK8 facilitates the translocation of STAT3 nuclear translocation and STAT3-dependent gene transcription. Its deficiency results in impaired Th17 differentiation from naïve CD4+ T cells, contributing to the AR hyper-IgE syndrome (HIES), or Job syndrome. Patients suffer from eczema-like rashes, recurrent skin infections, susceptibility to severe lung infections, and colitis.

Autosomal dominant HIES is linked to GOF mutations in the STAT3 gene. These patients present with lymphoproliferation, lymphadenopathy, recurrent staphylococcal abscesses or recurrent pneumonia, eczema, spontaneous bone fractures, and autoimmunity affecting multiple organs [23,24]. Elevated IgE and hyper eosinophilia are typical. In a study of 42 patients, Fabre A et al. reported that 24 had celiac-like enteropathy, and among them, five experienced colitis [123].

To date, no monogenic diseases specifically involving TH2 differentiation defects, such as GATA3 deficiency, have been identified. However, mutations in genes associated with TGF-β signaling pathways are linked to Loeys–Dietz syndrome, a complex connective tissue disorder characterized by aortic aneurysms, arterial tortuosity, hypertelorism, and a bifid or broad uvula or cleft palate. This syndrome is associated with several Mendelian diseases involving mutations in TGFBR1, TGFBR2, TGFB2, TGFB3, SMAD2, SMAD3, and IPO8 [124]. Patients exhibit an increased susceptibility to allergic diseases and IBD-like disease. Interestingly, TGFBR1/2-deficient individuals exhibit elevated TH2 cytokine production [22].

### 4.2. B Cells

IgA produced by intestinal B cells has different physiological functions compared to that of other B cells, such as not activating the complement cascade (Figure 1). An environment rich in anti-inflammatory cytokine TGF-β is required in Peyer’s patches for IgA class switching. Antibodies against Saccharomyces cerevisiae, Escherichia coli membrane protein C (OmpC), and bacterial flagellin (CBir1) are detected in IBD patients, though their involvement in pathogenesis is unclear. The pathophysiology of IBD involves a breakdown in the balance between inflammatory IgG [125,126] and anti-inflammatory IgA, leading to intestinal inflammation. IgA plays a role in microbiota diversity, while IgG anti-commensal antibodies are responsible for inflammatory disease in mice through macrophage IL-17 production [126]. Moreover, increased numbers of B cells have been reported in association with modifications in B cell methylation in IBD, which may alter levels of pro-inflammatory gene expression and contribute to disease activity [127]. Notably, several monogenic disorders characterized by altered B cell function and impaired immunoglobulin production are associated with disruptions in intestinal homeostasis, leading to inflammatory conditions. These include CVID, hyper-IgM syndrome, and agammaglobulinemia.

For example, lipopolysaccharide-responsive beige-like anchor (LRBA), a protein involved in the regulation of endosomal trafficking and implicated in type 8 CVID, leads to severe colitis and is characterized by increased apoptosis in CD19 B cells [128]. Hyper-IgM syndrome arises due to different gene mutations in CD40LG, AID (cytidine deaminase), or IKBKG. X-linked agammaglobulinemia, which arises due to Bruton’s tyrosine kinase (BTK) mutations in genes important for B cell development, is less frequently associated with colitis compared to CVID subsets and is characterized by less severe immunoglobulin deficiency [25].

## 5. Altered Regulatory Immunity in IBD and MID

Intestinal homeostasis is a complex balance between pro-inflammatory and anti-inflammatory factors. Immune regulation is a mandatory step in the immune response in the intestinal mucosa to avoid irreversible mucosal damage. In IBD patients, a quantitative and/or functional deficiency in regulatory populations could disrupt the balance between effectors and regulators, leading to the development of intestinal inflammation. The principal actors in immune regulation are regulatory T cells (Tregs), which express a specific intracellular transcription factor called forkhead box P3 (FOXP3). Tregs regulate intestinal inflammation through various mechanisms, including the production of IL-10 and TGF-β, the expression of cytotoxic T-lymphocyte antigen 4 (CTLA4), and the production of IL-2 and adenosine deaminase (ADA). Interestingly, through microbiota stimulation, Tregs express RORγt transcription factor, balancing the immune response in the GI mucosa [129]. Multiple mechanisms contribute to impaired T cell regulation in IBD [130]. One of the hypotheses is there is an inadequate number of functioning T cells, which has been demonstrated in animal induced-colitis models. X-linked defects in FOXP3 expression further support the critical role of Tregs in the maintenance of intestinal homeostasis. With more than 150 patients reported in the literature, X-linked defects in FOXP3 expression are among the most reported monogenic defects. Patients experience early severe inflammatory enterocolitis, early diabetes, severe inflammatory skin disorders such as eczema, and multiple organ autoimmunity [26]. As an X-linked disease, only boys are affected. FOXP3 mutations lead to a decrease in FOXP3 expression in T lymphocytes, resulting in a deficiency of Tregs.

For IBD, several studies have evaluated the number of Tregs in CD and UC with contradictory results from peripheral blood samples [131,132]. However, percentages of Foxp3+ T cells were reliably increased in the gut *lamina propria* from CD patients [133].

Defects in TReg functions may contribute to intestinal inflammation. IL-10 and TGF-β are two main immune regulatory cytokines with anti-inflammatory properties. IL-10 binds to IL-10RA and IL-10RB, allowing for the transduction of the JAK1/STAT3 pathway. This leads to the downregulation of Th1 cytokine expression and reduced expression of MHC II on macrophages. IL-10/IL-10R deficiency is the paradigm of IBD-like monogenic disease. Patients present with severe colitis associated with perianal lesions starting in the first year of life [27]. Additionally, patients experience extraintestinal diseases characterized by folliculitis, arthritis, and respiratory diseases [134]. Hematopoietic stem cell transplantation (HSCT) is the only curative treatment so far. Biallelic LOF variants in TGFβ1 were described in 2018 [135,136]. Patients present with a complex disease characterized by VEOIBD and central nervous system (CNS) disease, including epilepsy, brain atrophy, and posterior leukoencephalopathy.

Other monogenic diseases are responsible for altered Treg functions, such as CD25 deficiency or autosomal dominant CTLA4 mutations. CTLA4 is a membrane receptor that competes with the co-stimulatory protein CD28 for binding to CD80/CD86 and is involved in decreasing lymphocyte activation. A defect in CTLA4 leads to autoimmune cytopenia, IBD, psoriasis, and thyroid impairment. Patients also develop splenomegaly, hepatomegaly, bronchiectasis, granulomatous lymphocytic interstitial lung disease (GLILD), and generalized lymphadenopathy [28]. Patients can be treated with abatacept, a CTLA-4-immunoglobulin fusion.

## 6. Common Therapeutic Approach in IBD and MID

Therapeutic strategies in MID presents significant challenges due to the lack of personalized therapeutic approaches. The diagnosis of patient-specific genetic defects is crucial, as it enables a more tailored therapeutic approach. For example, patients presenting with pathogenic variants in the JAK/Stat pathway such as PTPN2, STAT3 GOF, or SOCS1 deficiency may benefit from JAK inhibitors (e.g., Tofacitinib or rutxolitinib) [137,138]. Interestingly, JAK inhibitors have demonstrated efficacy in treating IBD patients that are resistant to anti-TNFa therapy. In particular, STAT3 and STAT1 GOF, which lead to excessive immune activation, have further substantiated the rationale for the use of JAK inhibitors in such patients (e.g., Tofacitinib or rutxolitinib) [137,138]. Recent prospective studies in UC have confirmed the efficacy of anti-Jack inhibitors in patients resistant to anti-TNFa therapy. Upadacitinib (ABT 494), a putatively selective JAK1/2 inhibitor, has shown consistent therapeutic benefit across multiple inflammatory conditions including RA, PsA, AS, JIA, and IBD.

The role of regulatory T (Treg) cells has also been extensively studied in the context of both MID and IBD. Disorders such as IPEX and IPEX-like syndromes—resulting from mutations in FOXP3, CTLA4, and IL2RA (CD25)—underscore the essential role of immune regulation in preserving intestinal homeostasis. Therapeutic strategies targeting Treg cells are currently under active investigation, with more than 50 clinical trials—either ongoing or completed—registered on ClinicalTrials.gov. Many of these focus on adoptive Treg cell transfer as a promising therapeutic intervention.

In contrast, some drugs that are effective in MID have shown limited or no benefit in IBD. For example, abatacept, a CTLA-4-Ig fusion protein that inhibits the CD80/CD86 co-stimulatory signal required for T cell activation, has proven effective in patients with CTLA4 deficiency. However, this therapeutic approach has failed to demonstrate significant clinical efficacy in IBD patients.

Further insights into the pathophysiology of chronic granulomatous disease (CGD)—particularly its underlying reactive oxygen species (ROS) dysregulation—may support the development of novel therapeutic strategies for IBD. Targeting ROS pathways holds potential as an adjunctive or alternative treatment modality. Currently, a wide range of therapeutic options are available for IBD patients; however, heterogeneity in patient responses remains a major challenge. What proves effective for one patient may be ineffective or even harmful in another. In this context, precision medicine offers a promising avenue. Advanced methodologies—such as metagenomics, host genetic profiling, and gut microbiota sequencing—could enable more refined patient stratification. Specifically, a detailed characterization of individuals with MIDs using these precision tools may provide a framework for classifying IBD patients according to dominant immune signatures.

This immune-based stratification could facilitate the selection of targeted therapies tailored to each patient’s specific immunological profile. In 2022, Boston et al. conducted a multidimensional classification of monogenic IBD, integrating clinical phenotyping, single-cell RNA sequencing of intestinal biopsies, and genomic analysis [139]. Their findings revealed convergent cellular pathways between monogenic and polygenic IBD, suggesting that insights from rare monogenic diseases can inform therapeutic approaches in more common IBD forms.

## 7. Conclusions and Perspectives

The maintenance of intestinal homeostasis depends on a finely regulated interplay between epithelial barrier integrity, mucus secretion, and immune cell modulation within the lamina propria. This balance between immune tolerance and inflammation is crucial for maintaining gastrointestinal health. Although significant progress has been made in understanding the underlying mechanisms of intestinal inflammation, effective treatment remains elusive for many patients.

Recent therapeutic innovations—such as IL-23 inhibitors and JAK inhibitors—have expanded the treatment landscape for IBD. Nevertheless, a substantial proportion of patients remain unresponsive to current therapies. Consequently, a key objective for future research is the identification of predictive biomarkers that can inform therapeutic decision making and personalize treatment regimens.

The study of monogenic intestinal diseases provides a unique window into immune pathways relevant to IBD. For example, IL-18 has emerged as a key cytokine implicated in intestinal inflammation, particularly in conditions such as NLRC4 GOF mutations and XIAP deficiency. Targeting IL-18 may represent a novel therapeutic avenue for IBD, given its pivotal role in maintaining mucosal immune equilibrium.

Moreover, research into epithelial structural defects is gaining momentum. For example, Izumi Kaji et al. demonstrated the beneficial role of lysophosphatidic acid (LPA) signaling in restoring epithelial cell function [140]. Although not traditionally classified as an immune-targeted therapy, approaches that enhance epithelial barrier function may offer synergistic benefits when combined with anti-inflammatory treatments in IBD.

In conclusion, the integration of insights from monogenic disorders, molecular profiling, and advanced immunological techniques promises to reshape the future of IBD treatment—moving toward a more precise, personalized, and effective therapeutic paradigm.

## Figures and Tables

**Figure 1 ijms-26-06133-f001:**
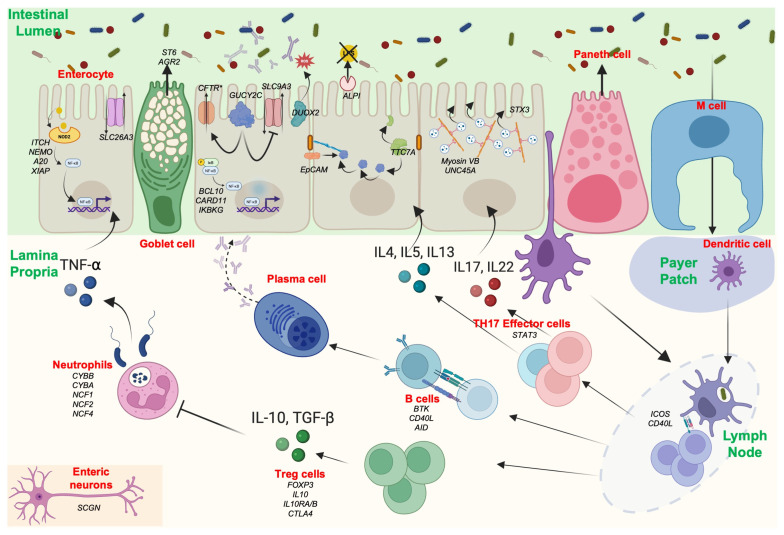
Mutated genes implicated in gastrointestinal homeostasis.

**Figure 2 ijms-26-06133-f002:**
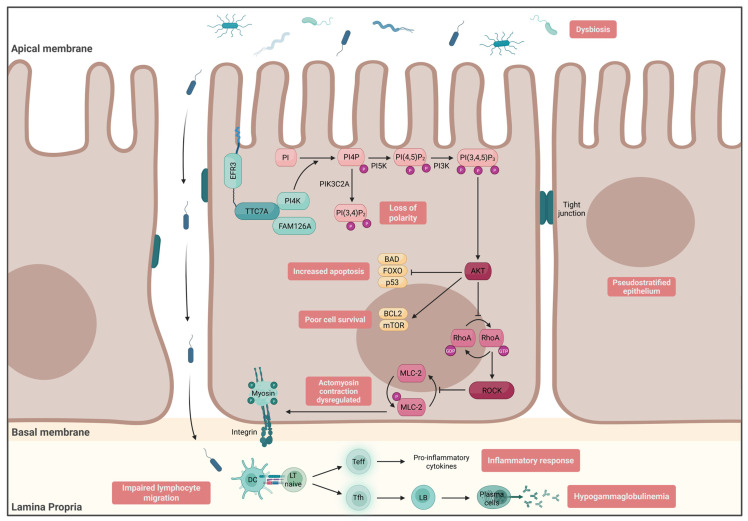
The role of TTC7A in intestinal homeostasis and the pathophysiological impact of its deficiency.

**Figure 3 ijms-26-06133-f003:**
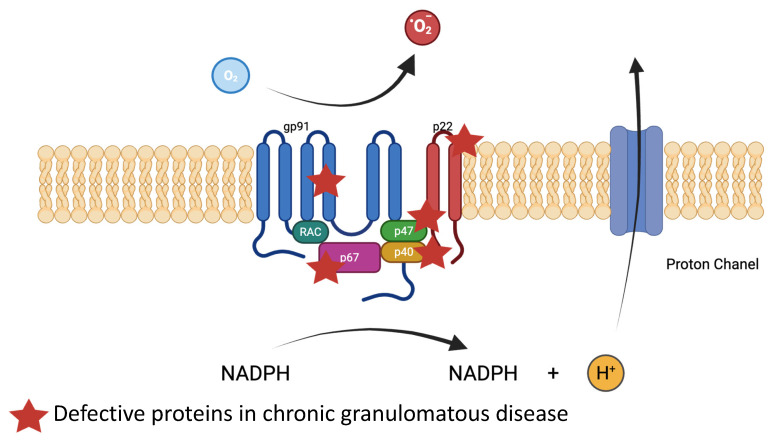
NADPH oxidase proteins and mutations linked to chronic granulomatous disease.

**Figure 4 ijms-26-06133-f004:**
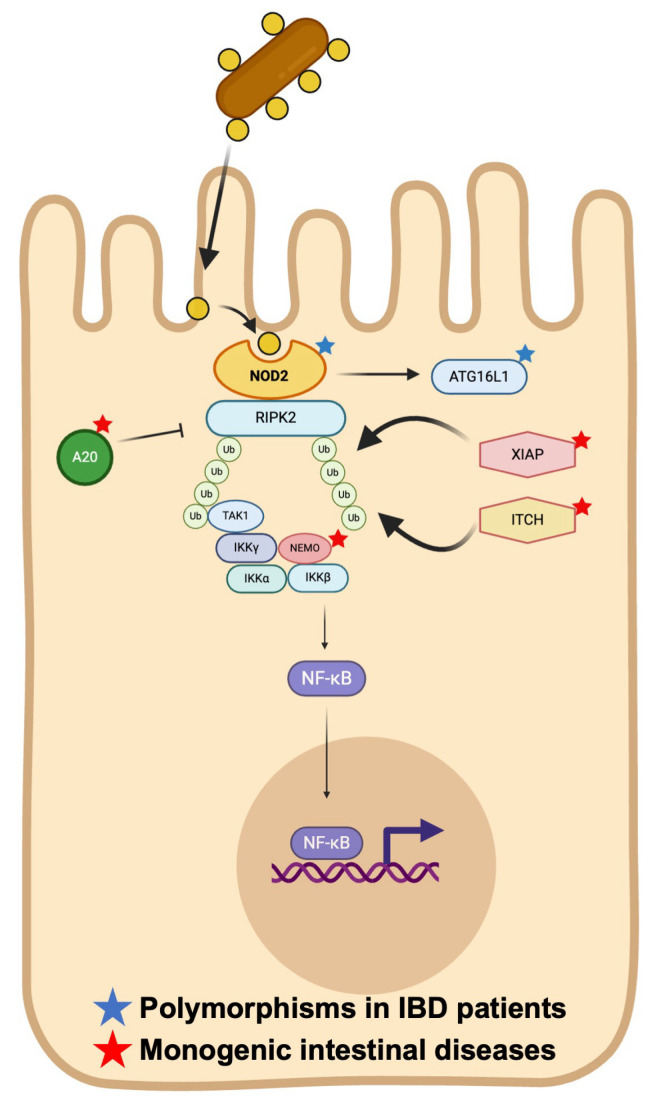
Key partners in the NOD2/CARD15 signaling pathway.

**Table 1 ijms-26-06133-t001:** Common Pathways Involved in Inflammatory Bowel Disease and Monogenic Intestinal Diseases.

IBD Pathophysiology	Genes Related to Monogenic Disease
**Gastrointestinal barrier**	
- Enterocytes	
Tight junctions Transport defect Enterocyte architecture	*EpCAM* [5] *SLC26A3* [6], *GUCY2C* [7,8,9] *MYO5B* [10], *TTC7A* [11]
- Mucus and goblet cells	*ST6* [12], *AGR2* [13]
**Innate Immunity**	
- Innate immunity in epithelial cells	*ALPI* [14], *DUOX2* [15]
- Innate immunity in immune cells	*CYBB*, *CYBA*, *NCF1*, *NCF2*, *NCF4* [16]
- Innate Immunity in epithelial cells and immune cells	*ITCH* [17], *NEMO* [18], *A20* [19]
**Adaptive immunity**	
- Effector T cells	*ICOS* [20], *DOCK8* [21], *TGFbR1/2/3*, *SMAD2/3*, *IPO8* [22], *STAT3* [23,24]
- B cells	*BTK, CD40L*, *AID* [25]
**Immune regulation**	
- Regulatory T cells	*FOXP3* [26], *IL10*, *IL10RA/B* [27], *CTLA4* [28]

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
