# Peer review of "Advances in Understanding Intestinal Homeostasis: Lessons from Inflammatory Bowel Disease and Monogenic Intestinal Disorder Pathogenesis"

_ijms, 2025, doi:10.3390/ijms26136133_

Round 1
Reviewer 1 Report
Comments and Suggestions for Authors
The authors provide a comprehensive overview of intestinal homeostasis in Irritable Bowel Disease (IBD) and Monogenic Intestinal Disorders (MID). This review examines altered conditions such as disruptions in gastrointestinal barriers, microbiota-host mucosal interactions, and immune system dysregulation that contribute to these diseases. The introduction and the beginning of the second chapter are appropriately detailed, effectively summarizing the literature while offering sufficient background to clarify lesser-known terms. Subsequently, the authors discuss the relationship between the host and gut microbiome, the immunological changes associated with intestinal diseases, and the genetic factors underlying these changes in IBD and MID. The study presents numerous genetic polymorphisms, supported by extensive references that effectively illustrate their relevance. Additionally, the review describes various genetic defects that contribute to intestinal diseases, offering a thorough and well-structured analysis. Overall, the review is well-written, logically organized, and makes a significant contribution to the field. I only noticed a few minor editorial issues. The figures could be larger, as I had to zoom in considerably to read the text. Additionally, the font in the last paragraphs of chapters 3 and 5 appears smaller than the rest of the text. In conclusion, this review is a valuable and well-executed work that significantly enhances the literature on this topic. I commend the authors for their excellent contribution.
Author Response
We thank Reviewer 1 for their comments. We have increased the size of the figures and harmonized the fonts across the different chapters.
Reviewer 2 Report
Comments and Suggestions for Authors
This is an interesting and exhaustive review that highlights the connection between monogenic intestinal diseases (MID) and potential common pathways involved in IBD pathogenesis. Understanding MID (monogenic intestinal diseases) which are driven by mutations in a single gene, begin early in life in children, result from marked disruptions of intestinal homeostasis and present more severe symptoms than IBD will be helpful in elucidating mechanisms underlying severe intestinal homeostasis disruption. The review further emphasizes that due to the proximity of MID and IBD in terms of phenotype and pathophysiological pathways, identifying and applying new omics methods to MID could be beneficial and can help physicians with more options of personalized treatments for children with severe intestinal homeostasis disruption and still having a negative genetic diagnosis. There were no major weaknesses as the review was on point. However, the authors should consider the following minor concerns:
- It is well known that in the human intestine, SLC26A3 and SLC9A3 are coupled and play an important role in electroneutral NaCl absorption. In addition to the role of SLC26A3 gene (associated with MID) in transport defect, is SLC9A3 gene also associated with MID and transport defect? There is a study that does mention about its association with VEOIBD (PMID:31833544). It would be of interest if the authors can clarify/discuss on this aspect.
- It will be helpful to the readers if the authors can provide a List of Abbreviations.
Author Response
We sincerely thank the reviewer for their thoughtful and constructive feedback on our manuscript. We appreciate the time and effort invested in reviewing our work, and we have carefully considered each comment. Below, we provide detailed responses to the reviewer’s points and describe the corresponding changes made in the revised manuscript.
This is an interesting and exhaustive review that highlights the connection between monogenic intestinal diseases (MID) and potential common pathways involved in IBD pathogenesis. Understanding MID (monogenic intestinal diseases) which are driven by mutations in a single gene, begin early in life in children, result from marked disruptions of intestinal homeostasis and present more severe symptoms than IBD will be helpful in elucidating mechanisms underlying severe intestinal homeostasis disruption. The review further emphasizes that due to the proximity of MID and IBD in terms of phenotype and pathophysiological pathways, identifying and applying new omics methods to MID could be beneficial and can help physicians with more options of personalized treatments for children with severe intestinal homeostasis disruption and still having a negative genetic diagnosis. There were no major weaknesses as the review was on point. However, the authors should consider the following minor concerns:
Q1. It is well known that in the human intestine, SLC26A3 and SLC9A3 are coupled and play an important role in electroneutral NaCl absorption. In addition to the role of SLC26A3 gene (associated with MID) in transport defect, is SLC9A3 gene also associated with MID and transport defect? There is a study that does mention about its association with VEOIBD (PMID:31833544). It would be of interest if the authors can clarify/discuss on this aspect.
A1. We thank the reviewer for bringing this important point to our attention. In response, we have expanded the discussion in the section on transport defects to include the potential involvement of the SLC9A3 gene. We now cite the study (PMID:31833544) and briefly discuss its reported association with very early onset IBD (VEOIBD), highlighting its relevance to monogenic intestinal disorders and its potential contribution to transport abnormalities in the context of MID.
Q2. It will be helpful to the readers if the authors can provide a List of Abbreviations.
A2. We appreciate the reviewer’s suggestion. Accordingly, we have added a glossary of abbreviations at the beginning of the manuscript to enhance clarity and facilitate the reader’s understanding.
Reviewer 3 Report
Comments and Suggestions for Authors
This review gives a good overview of the mechanisms behind intestinal homeostasis, particularly the focus on inflammatory bowel disease (IBD) and monogenic intestinal disorders (MID). The authors do a nice job of showing how genetic predisposition, immune regulation, and environmental factors work together in these conditions. The use of next-generation sequencing (NGS) and the study of monogenic defects provide useful insights into the complexity of IBD. However, this manuscript could be improved in some areas, in order to make it clearer and more impactful. Here are my specific comments:
1. The introduction could be more concise in summarizing the main goals and contributions of the review. This would help readers understand the focus and importance of the work better.
2. The discussion of IBD and MID is thorough, but the author should highlight the practical applications of these findings. For example, how can perception from monogenic disorders aid the development of personalized treatments for IBD?
3. The section on the gastrointestinal barrier is well-written, but it could be better if the authors include more recent studies or new ideas, such as the gut-brain axis or effect to barrier function from diet.
4. The review would benefit from critically include the limitations of current and ongoing research. For example, what are the challenges in applying findings from monogenic disorders to complex diseases like IBD?
5. The author should consider adding more visualization, such as diagrams or tables summarizing key pathways or gene interactions. I found Figure 1 is helpful but it would be better to expand it more to show complexity.
6. The discussion part is informative but is still quite limited. It would be good to include more details about ongoing clinical trials or recent therapies, particularly the targeting pathways that identified through monogenic studies.
7. The author could better address the differences in IBD and MID. For example, how do variations in disease presentation or patient demographics affect the interpretation of findings from monogenic disorders?
8. The author can consider provide deeper discussion of the role of the gut microbiota in both IBD and MID. Since the information is briefly mentioned, a more detailed exploration of how microbial dysbiosis interacts with genetic and immune factors would be very beneficial.
9. I think the conclusion feels a bit abrupt. It could be expanded to better summarize the key points and future directions. For example, the authors can highlight unanswered questions or areas for future research, so this would make an ending looks stronger.
10. More recent/ updated studies can be included, especially from the last two years. This would ensure the review reflects the latest advancements in the field.
Comments on the Quality of English Languagenone
Author Response
We sincerely thank the reviewer for their thoughtful and constructive feedback on our manuscript. We appreciate the time and effort invested in reviewing our work, and we have carefully considered each comment. Below, we provide detailed responses to the reviewer’s points and describe the corresponding changes made in the revised manuscript.
This review gives a good overview of the mechanisms behind intestinal homeostasis, particularly the focus on inflammatory bowel disease (IBD) and monogenic intestinal disorders (MID). The authors do a nice job of showing how genetic predisposition, immune regulation, and environmental factors work together in these conditions. The use of next-generation sequencing (NGS) and the study of monogenic defects provide useful insights into the complexity of IBD. However, this manuscript could be improved in some areas, in order to make it clearer and more impactful. Here are my specific comments:
Q1. The introduction could be more concise in summarizing the main goals and contributions of the review. This would help readers understand the focus and importance of the work better.
A.1 We have revised the introduction to more clearly and concisely summarize the main goals and key contributions of the review. We believe this improves the clarity and focus of the manuscript, and better highlights the relevance and significance of our work.
Q2. The discussion of IBD and MID is thorough, but the author should highlight the practical applications of these findings. For example, how can perception from monogenic disorders aid the development of personalized treatments for IBD?
A.2 We have added a dedicated paragraph in the discussion section that highlights the therapeutic implications of our findings. Specifically, we discuss how insights from monogenic immune disorders can inform the development of personalized treatment strategies for IBD, including the identification of molecular targets and the potential for stratified therapeutic approaches.
Q3. The section on the gastrointestinal barrier is well-written, but it could be better if the authors include more recent studies or new ideas, such as the gut-brain axis or effect to barrier function from diet.
A.3 We thank the reviewer for this constructive suggestion. In response, we have expanded the section entitled “Microbiota in IBD and MID” to incorporate recent findings and emerging concepts, including the gut-brain axis and dietary influences on barrier function. Specifically, we added the example of SCGN (secretagogin) deficiency, which has been shown to increase susceptibility to colitis and illustrates the relevance of neuro-immune interactions in intestinal homeostasis. We believe these additions enhance the depth and contemporary relevance of the discussion.
Q4. The review would benefit from critically include the limitations of current and ongoing research. For example, what are the challenges in applying findings from monogenic disorders to complex diseases like IBD?
A4. We have addressed this point in the newly added therapeutic implications paragraph.
Q5. The author should consider adding more visualization, such as diagrams or tables summarizing key pathways or gene interactions. I found Figure 1 is helpful but it would be better to expand it more to show complexity.
A.5 we have revised Figure 1 to include more detailed and complex representations of cellular interactions relevant to the discussed pathways. Additionally, we have included a new figure illustrating the mechanisms and pathophysiology of TTC7A deficiency, which we believe further enhances the clarity and educational value of the manuscript.
Q6. The discussion part is informative but is still quite limited. It would be good to include more details about ongoing clinical trials or recent therapies, particularly the targeting pathways that identified through monogenic studies.
A.6 We have revised the conclusion section to include a broader discussion of recent therapeutic developments and ongoing clinical trials informed by monogenic research. Specifically, we highlight emerging approaches such as anti-IL-18 therapy and recent studies involving lysophosphatidic acid (LPA) in MYO5B-deficient models. These additions aim to underscore the translational potential of monogenic findings and provide a more comprehensive outlook on future therapeutic directions.
Q.7. The author could better address the differences in IBD and MID. For example, how do variations in disease presentation or patient demographics affect the interpretation of findings from monogenic disorders?
A.7. Thank you for your comment and for pointing out the importance of addressing the differences between inflammatory bowel disease (IBD) and monogenic intestinal disorders (MID), particularly regarding disease presentation and patient demographics. Our review is primarily focused on the mechanistic roles of gene mutations and their impact on intestinal homeostasis, rather than on the clinical distinctions between IBD and MID. While we recognize the relevance of clinical variability, our aim was to emphasize the underlying biological mechanisms shared across different monogenic conditions. To provide clinical context, we have integrated key clinical features within each disease-specific section where possible.
Q.8. The author can consider provide deeper discussion of the role of the gut microbiota in both IBD and MID. Since the information is briefly mentioned, a more detailed exploration of how microbial dysbiosis interacts with genetic and immune factors would be very beneficial.
A.8 we have added a dedicated paragraph summarizing key studies investigating the gut microbiota in patients with MID.
Q.9. I think the conclusion feels a bit abrupt. It could be expanded to better summarize the key points and future directions. For example, the authors can highlight unanswered questions or areas for future research, so this would make an ending looks stronger.
A.9. we have revised the conclusion to provide a clearer summary of the main findings and insights presented in the review. At the beginning of the conclusion, we now summarize the key points, and we have expanded the final section to highlight emerging therapeutic targets and identify areas where further research is needed.
Q.10. More recent/ updated studies can be included, especially from the last two years. This would ensure the review reflects the latest advancements in the field.
A.10. We added new recent publications